# Early Bolting, Yield, and Quality of *Angelica sinensis* (Oliv.) Diels Responses to Intercropping Patterns

**DOI:** 10.3390/plants11212950

**Published:** 2022-11-01

**Authors:** Lucun Yang, Jingjing Li, Yuanming Xiao, Guoying Zhou

**Affiliations:** 1Northwest Institute of Plateau Biology, Chinese Academy of Sciences, Xining 810008, China; 2The Key Laboratory of Adaptation and Evolution of Plateau Biota, Chinese Academy of Sciences, Xining 810008, China; 3Academy of Animal Science and Veterinary Medicine, Qinghai University, Xining 810016, China

**Keywords:** *Vicia faba*, total land productivity, sole cropping of *Angelica sinensis*

## Abstract

Intercropping is a sustainable method for cultivating medicinal herbs since it requires lower dependence on chemical fertilizers than a sole cropping system. In this study, we compared the effects of sole cropping and intercropping on early bolting, yield, and the chemical composition of *Angelica sinensis* (Oliv.) Diels. Field experiments were conducted, in 2018 and in 2019, using different cropping systems including sole cropping of *A. sinensis* (AS), sole cropping of *Vicia faba* (VF), and intercropping (without fertilization) at three ratios: one row of *A. sinensis* + three rows of *V. faba*, AS/VF (1:3), two rows of *A. sinensis* + two rows *V. faba*, AS/VF (2:2), three rows of *A. sinensis* + one row *V. faba*, AS/VF (3:1). The effect of each cropping system was evaluated by measuring the dry biomass of *V. faba* and the dry biomass, ferulic acid content, and essential oil content and composition of *A. sinensis*. The early bolting rate of *A. sinensis* was significantly lower in the intercropping system as compared with that in a sole cropping system. The AS/VF (3:1) intercropping pattern resulted in an optimal yield and the highest ferulic acid content of *A. sinensis*, highest dry biomass of *V. faba,* and highest land equivalent ratio (LER). Additionally, the *A. sinensis* was more aggressive (the aggressivity value of *A. sinensis* was positive, and its competitive ratio was >1) under AS/VF (3:1) intercropping pattern, and it dominated over *V. faba* (which had negative aggressivity values and a competitive ratio of <1) under AS/VF (3:1) intercropping pattern. Ligustilide was the most dominant component of the essential oil of *A. sinensis*, regardless of the cropping system; however, the chemical component of essential oil was not influenced by intercropping patterns. Overall, the AS/VF (3:1) intercropping pattern without fertilization was the most productive, with the highest LER and ferulic acid content. These data indicate that intercropping can serve as an alternative for reducing the use of chemical fertilizers and intercropping also decreases the early bolting rate of *A. sinensis*, thus, enabling its sustainable production.

## 1. Introduction

*Angelica sinensis* (Oliv.) Diels (Apiaceae) is a biennial medicinal herb that grows in shrub meadows, forest margins, hillsides, and river valleys in China [1]. As a famous traditional Chinese medicine, the root of *A. sinensis* has been used for thousands of years. In the past, *A. sinensis* has been traditionally prescribed for treating gynecological diseases because of its ability to “replenish and invigorate the blood” [1,2]. In recent years, in-depth research on the chemical constituents and pharmacological action of *A. sinensis* has prompted its use for managing cancer [3], cardiovascular disease [4], and Alzheimer’s disease [5]. Meanwhile, it has been used as an ingredient in health care products and cosmetics [6]. Therefore, the increase in application range has led to an increased demand for *A*. *sinensis*. In past years, exploitation and habitat destruction have seriously threatened the resources of all natural production areas of *A*. *sinensis*. Investigations of wild *A. sinensis* have revealed that the survival of *A. sinensis* in many areas of China has been endangered because of habit fragmentation and excessive exploitation [7,8]. Large-scale cultivation of *A. sinensis* would ensure its conservation in natural stands and would also help to meet the ever-increasing market demand for its roots that possess medicinal properties. Recently, significant efforts have been made in the main production areas of *A. sinensis* to enhance its yield and quality [9,10,11], and these efforts have mainly been focused on different fertilization treatments [9,11,12]. However, excessive use of chemical fertilizers during *A. sinensis* cultivation often results in contamination of its roots, which generates human health concerns [13]. In addition, early bolting of *A. sinensis* has become a major bottleneck in agricultural production. *A. sinensis* generally shows an early bolting rate of 30–50%, but it can increase up to 60–80% under certain conditions or stresses. During early bolting, large quantities of nutrients in the roots of *A. sinensis* are redirected to the reproductive organs for seed production. This leads to the lignification of roots and loss of their medicinal value, thus, severely restricting the yield and quality of *A. sinensis* [14]. This challenged us to find a technique that would reduce the amount of fertilizer input and would also reduce the early bolting rate of *A. sinensis.*

Intercropping is a sustainable farming practice where two or more crops can be grown simultaneously in the same field. As compared with an equivalent area under sole cropping, intercropping can enhance total land productivity because of effective utilization of available environmental resources such as light, nutrients, and water [15], thus, resulting in a higher land equivalent ratio (LER) [16]. Previous studies have shown that legume species play a vital role in an intercropping system, as they supply nitrogen to the companion crops by nitrogen fixation and also promote resource efficiency [17,18,19,20,21,22]. Therefore, legumes can be used as substitutes for chemical fertilizers in intercropping systems for sustainable production [23].

Wild *A. sinensis* growing in forests and alpine shrubs requires partial shade. Previous studies have shown that shading during the growing season reduces early bolting in *A. sinensis*, thus, improving its yield and quality [24,25]. *V. faba* plants are tall and heliophilous, whereas *A. sinensis* plants are dwarf and sciophilous. In an intercropping system, *V. faba* can provide shade and consequently simulate the natural growth of *A. sinensis*. In addition, legume species have been shown to enhance the biomass yield and quality of medicinal plants in intercropping systems [26,27,28,29,30]. Therefore, we speculate that intercropping *A. sinensis* with *V. faba* would reduce the amount of fertilizer input and would also reduce the early bolting rate of *A. sinensis.* However, little is known about the intercropping of *A. sinensis* with *V. faba*.

To date, more than 70 compounds have been isolated and identified from *A. sinensis* [31]. Among these compounds, essential oil and ferulic acid, which determine the pharmacological properties of *A. sinensis*, are used as markers for the quality assessment of the dried root of *A. sinensis* (Chinese Pharmacopoeia Commission 2020). A minimum of 0.4% essential oil content and 0.05% ferulic acid content in *A. sinensis* dried root is required for its use as a medicinal or cosmetic product (Chinese Pharmacopoeia Commission 2020). Phthalides are key active ingredients of *A. sinensis* essential oil. Different types of phthalides have been isolated from *A. sinensis* such as ligustilide, butylphthalide, senkyunolide A, senkyunolide H, senkyunolide I, butylidenephthalide, and levistolide A [32].

In intercropping systems, competition among plants plays an important role in affecting plant growth and yield [33]. When intraspecific competition is higher than interspecific competition and also when there are mutualistic relationships among intercropping components, intercropping systems can be advantageous [34,35]. Many indices have been used to evaluate potential advantages of intercropping systems and species interactions, such as land equivalent ratio (LER), relative crowding coefficient (RCC or K), aggressivity (A), and competitive ratio (CR). In different soybean/peppermint intercropping systems the abovementioned indices have been used to depict competition and economic advantage [33,36,37,38].

In this study, we evaluated the effects of intercropping *A. sinensis* with *V. faba* on the yield, early bolting rate and essential oil content, and the yield and composition of *A. sinensis*, and assessed the benefit of intercropping and sole cropping by comparing the LER, RCC, A, and CR values in these cropping systems. We hypothesized that intercropping *A. sinensis* with *V. faba* would: (1) provide partial shade to *A. sinensis* plants, and thus, decreasing the early bolting rate; (2) decrease the yield of *A. sinensis* because of no fertilizer application, (3) result in higher quality of *A. sinensis* than sole cropping.

## 2. Results

### 2.1. Soil Chemical Properties

#### 2.1.1. Effect of Cropping System on Organic Matter, N, P, and K Contents of the Soil

There was no significant effect of cropping system on organic matter and K contents both in 2018 and in 2019 (Table 1). The available N and available P were influenced by the cropping system in 2018. The available N content in AS/VF (2:2) was significantly higher than sole cropping of *A. sinensis*, while the available P content in AS/VF (3:1) was significantly higher than sole cropping of *A. sinensis* (Table 1). However, the cropping system had no significant effect on available N and available P in 2019. Meanwhile, there was no significant difference in soil chemical properties between 2018 and 2019 (Table 1).

#### 2.1.2. Correlations among Soil Nutrients and Plant Growth and Essential Oil Production

The results of Spearman’s correlation coefficient values among soil organic matter, and available N, P, and K and biomass yield, ferulic acid content, essential oil content, and essential oil yield are presented in Figure 1. On the one hand, the results showed that the content of available K was negatively correlated with biomass yield, essential oil content, essential oil yield, and chloroplast content, both in 2018 and in 2019. On the other hand, in 2018, there was a significant negative correlation between biomass yield, essential oil yield, chloroplast content, and available N, while there was a significant positive correlation between available P and available N (Figure 1). These results for 2019 were the same as for 2018, except for the significant positive correlation between soil organic carbon and available N. In addition, biomass yield, essential oil content, essential oil yield, and ferulic acid content of *A. sinensis* increased with an increase in chlorophyll content, both in 2018 and in 2019.

### 2.2. Early Bolting Rate of A. sinensis

Intercropping significantly decreased the early bolting rate of *A. sinensis* as compared with the sole cropping system in 2018 and in 2019 (*p* < 0.05) (Table 2). In 2018, the early bolting rate of *A. sinensis* in the AS/VF (3:1), AS/VF (2:2), and AS/VF (1:3) intercropping patterns were 17.37%, 22.58%, and 26.30%, respectively, which were lower than that of the sole cropping system (36.78%) by 52.77%, 38.60%, and 29.22%, respectively. In 2019, the early bolting followed a similar pattern as in 2018. However, the early bolting rate of *A. sinensis* in 2018 was slightly higher than that in 2019.

### 2.3. Chlorophyll Content

The chlorophyll content of *V. faba* was significantly affected by the intercropping pattern in both years (Table 3). As compared with *V. faba* monoculture, the AS/VF (3:1) and AS/VF (2:2) intercropping patterns significantly increased the chlorophyll content in the leaves of *V. faba* in both years. In *A. sinensis*, although the chlorophyll content was decreased by intercropping, the magnitude of reduction was not significant in both years (*p* > 0.05) (Table 3). In addition, there was no significant difference of the chlorophyll contents in leaves of *V. faba* and *A. sinensis* between 2018 and 2019.

### 2.4. A. sinensis Root Biomass and V. faba Biomass Yield

The results demonstrated that there was a significant difference in *A. sinensis* root biomass and *V. faba* biomass yield under different cropping systems (Table 3). In 2018 and in 2019, the highest biomass yields (5042.8 kg/hm^2^ and 5116.5 kg/hm^2^) of *A. sinensis* were obtained under sole cropping system. As the number of planting rows of *A. sinensis* decreased, the biomass yield of *A. sinensis* also decreased. The lowest biomass yields (525.00 kg/hm^2^ and 638.5 kg/hm^2^) were acquired in the AS/VF (1:3) intercropping pattern in 2018 and in 2019, respectively (Table 3). Intercropping significantly enhanced the *V. faba* biomass yield. In 2018 and in 2019, the highest (10763.5 kg/hm^2^ and 11110.8 kg/hm^2^) and lowest (4319.6 kg/hm^2^ and 4433.3 kg/hm^2^) *V. faba* biomass yield were obtained under the AS/VF (3:1) intercropping system and the sole cropping system, respectively (Table 3). The results also indicated that there was no significant difference in the biomass yield of *A. sinensis* and *V. faba* biomass yield between 2018 and 2019.

### 2.5. Ferulic Acid Content

The ferulic acid content of *A. sinensis* roots was analyzed using HPLC (Appendix A). The results in relation to the ferulic acid content demonstrated significant differences among cropping systems in both years. On the one hand, the highest ferulic acid contents (0.16% and 0.14%) were recorded under the AS/VF (3:1) intercropping system which demonstrated about 13.22% and 8.07% increase over the sole cropping of *A. sinensis* in 2018 and in 2019, respectively (Table 4). On the other hand, the lowest ferulic acid contents (0.10% and 0.096%) were found in samples under the AS/VF (1:3) intercropping system in 2018 and in 2019, respectively. Overall, the ferulic acid content had no significant difference in both years.

### 2.6. Essential Oil Content, Yield, and Composition

As compared with the sole cropping system, the essential oil content of *A. sinensis* was higher in the AS/VF (2:2) intercropping pattern and lower in the AS/VF (3:1) and AS/VF (1:3) patterns, however, the difference in the essential oil content between sole cropping and intercropping systems was not significant (*p* > 0.05) (Table 4). Similar results were obtained in both years. These data suggest that intercropping has no significant impact on the essential oil content of *A. sinensis*.

The essential oil yield of *A. sinensis* was significantly higher in the sole cropping system than in the intercropping system both in 2018 and in 2019 (Table 4), that is, intercropping reduced the essential oil yield of *A. sinensis*. Meanwhile, there was no significant difference in essential oil and essential oil yield of *A. sinensis* between 2018 and 2019.

Analysis of the composition of *A. sinensis* essential oil revealed that ligustilide was the most abundant ingredient, and the content of ligustilide was not affected by intercropping (*p* > 0.05) in 2018 and 2019 (Table 5). The amount of senkyunolide A and butylphthalide was lower in the intercropping systems than in the sole cropping system, but the difference was not significant (Table 5). Similar results were obtained in both 2018 and 2019. Meanwhile, as for as senkyunolide H, senkyunolide I, levistolide A and butylidenephthalide, cropping system has no significant influence on them both in 2018 and in 2019 (Table 5). In addition, there was no significant difference in the chemical composition of essential oil between 2018 and 2019.

### 2.7. Competition Indices under Different Intercropping Patterns

The value of LER was greater than one in the AS/VF (3:1) and AS/VF (2:2) intercropping patterns (Figure 2) both in 2018 and in 2019. The highest LER values (1.24 and 1.25) were obtained in the AS/VF (3:1) intercropping pattern in 2018 and in 2019. The partial LER of *A. sinensis* decreased as the proportion of *A. sinensis* decreased in the intercropping patterns, while the partial LER of *V. faba* increased first and then decreased as the proportion of *A. sinensis* increased in the intercropping patterns.

Among the three intercropping patterns, the aggressivity value for *A. sinensis* (Aa) under the AS/VF (3:1) intercropping pattern was positive (2.319), while it was negative under the AS/VF (2:2) (−0.6298) and AS/VF (1:3) (−0.9178) intercropping patterns (Figure 3). The aggressivity value for *V. faba* (Av) was negative under the AS/VF (3:1) intercropping pattern, while it was positive under the AS/VF (2:2) and AS/VF (1:3) intercropping patterns. Over the two years, consistent observations were made. Among all the intercropping patterns, the competitive ability of *V. faba* was significantly improved, as indicated by greater values of RCCv as compared with the corresponding values of RCCa. In addition, the highest and lowest RCC values in 2018 were achieved in the intercropping patterns of AS/VF (3:1) and AS/VF (1:3), with RCC values of 21.18 and 1.31, respectively (Figure 4). The same was true for the RCC values in 2019. Among all the intercropping patterns, the partial CR of *A. sinensis* (CRa) was higher than one and higher than that of *V. faba* (CRv) (Figure 5) under the AS/VF (3:1) intercropping pattern. Furthermore, both in 2018 and in 2019, the highest and lowest CR values in *A. sinensis* were recorded under the AS/VF (3:1) and AS/VF (1:3) intercropping patterns, respectively.

## 3. Discussion

### 3.1. Early Bolting Rate of A. sinensis

In this study, intercropping significantly decreased the early bolting rate of *A. sinensis*, as expected. Previous studies have shown that early bolting in *A. sinensis* was affected by many factors, such as genetics, ecological conditions (light, climate, soil, and water), nutrients (organic acid, free amino acids, and soluble sugars), and cultivation practices (sowing date, harvest date, type and method of fertilization, and plant density). Lin (2010) [39] found that early bolting rate decreased by 5% and 5.5% when covered with 75% and 50% sun shade net, respectively. Han (2018) [9] showed that the ratio of organic fertilizer and chemical fertilizer applied to *A. sinensis* had a significant influence on early bolting; increasing the amount of organic fertilizer and decreasing the amount of chemical fertilizer significantly reduced early bolting. Liu (2003) [24] reported that photoperiod was potentially one of the main reasons affecting early bolting in *A. sinensis*. Intercropping of *V. faba* with *A. sinensis* reduced the natural light intensity through shading (sole crop of *A. sinensis*, 59.55 klx light intensity, AS/VF (3:1), 42.12 klx; AS/VF (2:2), 44.35 klx; AS/VF (1:3), 47.53 klx), which affected the photoperiod and consequently the early bolting rate of *A. sinensis*. Therefore, the low early bolting rate of *A. sinensis* in intercropping patterns observed in this study could be attributed to no fertilization and low light intensity due to shading by *V. faba* plants.

### 3.2. Dry Root Biomass of A. sinensis

In 2018 and in 2019, the highest dry root yields of *A. sinensis* were observed in the sole cropping system (5042.8 kg/hm^2^ and 5116.5 kg/hm^2^, respectively), followed by the AS/VF (3:1) intercropping pattern (4196.5 kg/hm^2^ and 4284.00 kg/hm^2^, respectively). Han (2018) [9] showed that the dry root yield of *A. sinensis* grown in the Gansu Province ranged from 2608.30 kg/hm^2^ (no fertilization) to 6250.00 kg/hm^2^ (A2B1C0: organic fertilizer, 36 kg; diammonium phosphate: 1.8 kg; and microbial fertilizer: 0 kg) under different fertilization treatments. Another study showed that the highest dry root yield of *A. sinensis* (3200 kg/hm^2^) was obtained in the pure organic fertilizer treatment in Gansu Province [11]. An experiment carried out at the same experimental facility in Qinghai Province showed that the dry root yield of *A. sinensis* ranged from 5749 to 7287.5 kg/hm^2^ under different fertilization treatments (Appendix A). In this study, the dry root yield of *A. sinensis* in the sole cropping system differed from that in the AS/VF (1:3) and AS/VF (2:2) intercropping patterns, and the dry root yield was not affected in the AS/VF (3:1) pattern, as the value was within the range reported previously. Additionally, previous studies have shown that an increase in the leaf chlorophyll content led to an increase in the photosynthetic activity and biomass of medicinal plants [40,41,42]. In this study, although the chlorophyll content of *A. sinensis* was not affected by the intercropping pattern, the increased trend of biomass was consistent with the increased trend of chlorophyll content, indicating that chlorophyll content was one of the factors affecting the biomass yield of *A. sinensis*. Therefore, higher *A. sinensis* biomass yield in the AS/VF (3:1) intercropping pattern may be due to the following reasons: (1) higher chlorophyll content of *A. sinensis*, leading to higher photosynthetic activity; (2) low early bolting rate; (3) efficient use of resources [43], due to the effect of *V. faba* on N availability for *A. sinensis*, or facilitative interaction between *V. faba* and *A. sinensis* plants in the intercropping system [44,45]. Moreover, water, radiation, nutrients, and other resources were probably more efficiently used because of differences in the morphological features and growth environments of intercropped plants [46,47]. Furthermore, intra- and interspecific competition was less in the AS/VF (3:1) intercropping pattern than in the sole cropping systems of *A. sinensis* and *V. faba*.

As compared with *A. sinensis* monoculture, the yield of *A. sinensis* per unit area in intercropping system decreased with a reduction in the proportion of *A. sinensis* plants. The results of this study are consistent with those of previous studies on the intercropping of dill/common bean [26], Moldavian/faba bean [27], black cumin/faba bean [28], and faba bean/peppermint [30].

### 3.3. Ferulic Acid Content

According to previous reports, the content of ferulic acid in *A. sinensis* ranges from 0.0211 to 0.143%, depending on the method of extraction [48,49,50,51,52,53,54]. In this study, the ferulic acid content of *A. sinensis* varied from 0.095 to 0.160%. In 2018 and in 2019, the highest ferulic acid contents (0.160% and 0.144%) were observed in the AS/VF (3:1) intercropping pattern, which were significantly higher than that obtained in the other two intercropping patterns. Additionally, the ferulic acid content obtained in the AS/VF (3:1) intercropping pattern was greater than the maximum values obtained in previous studies. Thus, the quality of *A. sinensis* was affected by the intercropping pattern, as the amount of ferulic acid in the AS/VF (3:1) intercropping pattern was higher than that reported previously. Several factors affect the biosynthesis of plant secondary metabolites, such as light, drought, mineral nutrients, and high salinity [55]. In previous studies, the contents of organic carbon (C), total N, total K, available K, calcium (Ca), copper (Cu), zinc (Zn), nickel (Ni), and manganese (Mn) in the soil were positively correlated with the ferulic acid content of *A. sinensis* [2]. The results in this study were inconsistent with those of previous studies, which showed that the ferulic acid content of *A. sinensis* was negatively correlated with the available K in the soil (Figure 1). Therefore, the reasons for the high ferulic acid in the AS/VF (3:1) intercropping pattern need further study.

### 3.4. Essential Oil Content, Yield, and Composition

The essential oil contents of *A. sinensis* roots ranged from 0.45 to 0.96% in this study, which was higher than 0.4% stipulated in the Chinese Pharmacopoeia Commission (2020). Rong et al. (2011) [56] showed that the essential oil content varied from 0.2152 to 1.6231% in different areas. Therefore, the essential oil contents of *A. sinensis* in different cropping systems in this study were within the range reported previously. The essential oil content of *A. sinensis* was not influenced by the intercropping pattern, which was supported by the results of Vafadar-Yengeje et al. (2019) [57], who reported no significant difference in the essential oil content of Moldavian balm between its monoculture and Moldavian balm/*V. faba* intercropping. It must be noted that N plays a vital role in determining the photosynthetic rate and the development and differentiation of essential oil-related cells [58,59]. Therefore, the reason why no difference was observed in the essential oil content of *A. sinensis* between intercropping and sole cropping systems was probably the absence of exogenous N supply [41].

Our finding that intercropping decreases the essential oil yield of *A. sinensis* was inconsistent with the results of Machiani et al. (2018) [30], Maffei and Mucciarelli (2003) [60], and Weisani et al. (2015) [26]; all three studies investigated legume/medicinal plant intercropping systems, which enhanced the essential oil yield of the medicinal plant as compared with its sole cropping system. Previous studies have shown that essential oil yield was influenced by crop biomass, biosynthesis rate, oil storage glands, and glandular trichome number and size [40]. In this study, the essential oil content of *A. sinensis* was not affected by intercropping, suggesting that the essential oil yield depends on the biomass yield of *A. sinensis*. This result was supported by a correlation analysis; the Spearman’s correlation coefficients of essential oil yield and biomass yield were significantly high (Figure 1), which demonstrated that any factor that enhanced the biomass yield could increase the essential oil yield [40]. Higher biomass yield is associated with higher levels of photosynthesis. Therefore, higher chlorophyll content of *A. sinensis* in the sole cropping system increased its level of photosynthesis, leading to higher essential oil yield [61].

In this study, ligustilide was identified as the principal component of the essential oil of *A. sinensis*, which was in agreement with previous reports [62,63,64,65]. Additionally, the content of ligustilide was not affected by intercropping (*p* > 0.05). However, the content of ligustilide in different cropping systems was higher than that in samples reported from previous studies [66,67], but comparable with a recent study using the same method as this study, reported by Gui and Zheng (2018) [65]. Although the contents of senkyunolide H and butylidenephthalide decreased in the intercropping system, their levels were within the range reported previously [65]. Hence, the chemical composition of *A. sinensis* essential oil was not influenced by the intercropping pattern. This further implied that the essential oil composition of *A. sinensis* obtained in the intercropping system without fertilization was comparable to that obtained in different fertilization treatments [55,63,65]. Some studies have shown that intercropping legumes with medicinal plants decreased the content of the main essential oil components and increased that of minor essential oil components [26,27], while other studies have indicated that intercropped plants contained a significantly higher content of most of the major essential oil components [28]. In addition, Maffei and Mucciarelli (2003) [60] reported that intercropping systems did not have a significant influence on the chemical profile of the essential oil, consistent with the current study.

### 3.5. Competition Indices

Values of the LER in the AS/VF (3:1) intercropping pattern in 2018 and in 2019 were 1.24 and 1.25, which implied that 24% and 25% more land area would be needed in the sole cropping system to reach the same yield as that obtained in the intercropping system, respectively [36]. These results suggest that intercropping offers a yield advantage over monoculture because of the efficient use of land and other environmental resources (solar radiation, nutrients, and water) for plant growth [15]. Our results were in agreement with the findings of Rostaei et al. (2018) [28], Fallah et al. (2018) [27], and Amani et al. (2018a) [29], who studied dill- faba bean, dragonhead/faba bean, and peppermint/faba bean intercropping, respectively.

Aggressivity was used to express how much higher the increase in *A. sinensis* production was as compared with *V. faba* in the intercropping system [33]. Considering all planting patterns, *A. sinensis* was the dominant species in the AS/VF (3:1) intercropping pattern. CR gives a better evaluation of the competitive ability among intercropping components and is more advantageous with respect to the aggressivity and RCC indices [33]. Generally, the CR index indicates the ratio of partial LER of two plants and takes into consideration the proportion of the plants with which they are initially sown [37]. The CR values of *A. sinensis* were less than one and higher than that of *V. faba* in the AS/VF (3:1) intercropping pattern, indicating an absolute yield advantage of *A. sinensis* over *V. faba* in intercropping patterns. This was due to the higher aggressivity of *A. sinensis* and its capability in optimizing the use of available resources as compared with *V. faba,* leading to a dominant position. Similar results were reported by Yilmaz et al. (2014) [36] in barley and vetch intercropping and by Lithourgidis et al. (2011) [33] in pea/cereal intercropping.

## 4. Materials and Methods

### 4.1. Plant Material

The *A. sinensis* seeds were obtained from the seed base of the Group for Medicinal Plant Resources and Vegetation Restoration of Tibetan Plateau of Northwest Institute of Plateau Biology, Chinese Academy of Sciences. A specimen with the voucher number Zhou2017115, identified by Professor Guoying Zhou, was collected from the seed base at the flowering stage. Seeds were sown in 2017 and in 2018, according to the standardized technical specifications for *A. sinensis*. Seedlings were extracted from the soil at 16 weeks after sowing, and then preserved.

### 4.2. Field Trial

The life cycle of *A. sinensis* is generally completed within 3 years (first year, seedling stage; second year, pharmaceutical period; and third year, bolting period). This study mainly focused on the pharmaceutical period. Field trials were conducted in the De Xing village, Beishan Township, Minghe County (102°47.70′E, 36°24.183′N, altitude of 2581 m above sea level), in 2018 and in 2019. The average monthly precipitation during both experimental seasons is illustrated in Figure 6. The weather data presented in Figure 6 revealed some heterogeneity in the average monthly precipitation between the two growing seasons. In 2018, precipitation was lower in April, May, and June than that in 2019, while precipitation was higher in July, August, September, and October than that in 2019. Soil characteristics, measured prior to the experiment using normal agrochemical procedures, were as follows: ripe soil type, yellow loam; pH = 7.5; organic matter = 12.73 g/kg; total nitrogen (N) = 1.49 g/kg; total phosphorus (P) = 0.83 g/kg; total potassium (K) = 19.27 g/kg; available N = 96.78 mg/kg; available P = 25.59 mg/kg; available K = 160 mg/kg. Prior to the experiment, potato (*Solanum tuberosum* L.) was grown on the test field.

The field experiment was carried out in a randomized block design, with one factor (cropping system) and three replications. Five treatments were performed in this experiment: sole cropping of *V. faba* (VF), sole cropping of *A. sinensis* (AS), and three intercropping ratios of *A. sinensis*/*V. faba*, i.e., one row of *A. sinensis* + three rows of *V. faba*, AS/VF (1:3); two rows of *A. sinensis* + two rows of *V. faba*, AS/VF (2:2); and three rows of *A. sinensis* + one row of *V. faba*, AS/VF (3:1). The soil was plowed to a depth of 30 cm, and the field was divided into blocks and plots (2.6 m × 5 m). The soil was covered with a black plastic sheet (1.2 m) before transplanting *A. sinensis* and *V. faba* seedlings. Each plot contained two columns and eight rows per column. *A. sinensis* seedlings of uniform size were transplanted into the experimental plots on 8 April 2018 and 12 April 2019. In both the sole cropping and intercropping plots, seedlings of *A. sinensis* were transplanted at 25 cm row-to-row spacing and 20 cm plant-to-plant spacing, whereas *V. faba* seeds were planted at 25 cm row-to-row spacing and 25 cm plant-to-plant spacing. No chemical fertilizers or pesticides were used during the experiment, and weeds were controlled manually. The experiment was carried out under natural conditions without additional irrigation.

### 4.3. Evaluation of the Early Bolting Rate of A. sinensis

*A. sinensis* plants showing 30 cm of the top of the extended tuft on the main stem were considered as early bolting plants. The number of early bolting plants was counted in each plot on 8 August 2018 and 15 August 2019. The early bolting rate (%) was calculated according to the following equation:

Early bolting rate (%) = (number of early bolting of *A. sinensis* plants in a plot)/(total number of *A. sinensis* plants in the plot) × 100%


### 4.4. Determination of V. faba Biomass Yield and A. sinensis Biomass Yield

The *A. sinensis* and *V. faba* plants were harvested from a 1 m^2^ area in each plot (with 100% planting density) at maturity (185 and 154 days after transplanting, respectively) in 2018 and in 2019. The root samples were meticulously washed with tap water, and then dried in the shade at room temperature for a month. The dried root samples of *A. sinensis* were weighed, and biomass yield (kg/hm^2^) was calculated. *V. faba* seeds were threshed and dried at 45 ℃ for 48 h, and *V. faba* dry biomass yield was calculated per unit area.

### 4.5. Chlorophyll Content Analysis

The chlorophyll contents of the *A. sinensis* and *V. faba* plants were evaluated using the SPAD meter (SPAD 502, Minolta Ltd., Osaka, Japan). Ten leaves were randomly selected per plot, and the middle section of the leaf blade was used for chlorophyll content measurements; the measurements were carried out at the same time (rapid growing stage) in 2018 and in 2019.

### 4.6. Soil Sampling and Analysis

After harvesting the *A. sinensis* plants, soil samples (0–30 cm depth) were collected and homogenized from each plot in 2018 and in 2019. Soil samples were sieved through a 2 mm mesh to remove any stones and roots, and then dried at room temperature. The air-dried soil samples were used for analysis to evaluate the chemical properties. The available N, available P, and available K contents were measured using the Kjeldahl method [68], Mo-VF colorimetric method [69], and ammonium acetate flame photometer method [69], respectively. The potassium dichromate oxidation spectrophotometric method was used to measure soil organic carbon [68].

### 4.7. Essential Oil Extraction

The dried roots of the *A. sinensis* plants were crushed and passed through a 60 mesh sieve. The ground root tissue (50 g) was added to a 1 L round-bottom flask containing 500 mL of distilled water. Then, the flask was connected to a Clevenger-type apparatus, as described by the Chinese Pharmacopoeia Commission (2020). Hydrodistillation was carried out for 5 h in nine replicates (*n* = 9), and the essential oil content (% (*v*/*w*)) and yield (gm^−2^) were calculated using the following equations:Essential oil(%) = Amount of essential oil extracted Amount of ground dried root × 100%
Essential oil yield = *A. sinensis* biomass yield × Percent essential oil content

### 4.8. Essential Oil Composition Analysis

#### 4.8.1. Sample Preparation

Fresh root samples of *A. sinensis* (1 g) were ground in liquid nitrogen. Then, for each sample, 1 mL of 70% methanol was added to the ground sample and vortexed for 3 min. The sample was shaken every six times at 30 min intervals and stored at 4 °C overnight. After overnight incubation, the sample was centrifuged at 12,000 rpm for 15 min, and 300 μL of the supernatant was filtered through a 0.2 μm filter and evaporated to dryness. The lyophilized samples were redissolved in 1 mL of water/methanol (1:1 [*v*/*v*]) solution, and centrifuged. The supernatant was collected and analyzed by liquid chromatography-tandem mass spectrometry (LS-MS/MS).

#### 4.8.2. LC-MS/MS Analysis

To perform the LC-MS/MS analysis, reference standards including levistolide A (CAS NO: 551-08-6), ligustilide (CAS NO: 4431-01-0), butylidenephthalide (CAS NO: 551-08-6), butylphthalide (CAS NO: 6066-49-5), senkyunolide I (CAS NO: 94596-28-8), senkyunolide H (CAS NO: 94596-27-7), and senkyunolide A (CAS NO: 63038-10-8) were purchased from Chengdu Balens Biotechnology Co. Ltd. (Sichuan, China) and stored in the dark at −20 °C. Acetonitrile and methanol (HPLC grade) were purchased from Tian in Fuyu Fine Chemical Co, Ltd. (Tianjing, China). All other reagents and chemicals (analytical grade) were purchased from Shandong Yu Wang He Tianxia New Material Co. Ltd. (Shandong, China).

The supernatant was separated using an analytical column (Waters ACQUITY UPLC HSS T3, USA), with a particle size of 1.8 μm and column size of 3.0 mm × 100 mm. The column temperate was maintained at 40 °C during separation. Solvent A (0.1% formic acid in water) and solvent B (0.1% formic acid in acetonitrile) were used as the mobile phases. A 3 μL injection volume was used, and the flow rate was maintained at 0.5 mL/min. Sample separation was carried out using the following gradient conditions: 0.5 min, 10% B; 5 min, 90% B; 5.1 min, 10% B; 7 min stop.

The mass spectra were acquired using the QTRAP4500 system with a Duo Spray source in positive and negative ESI modes using the following parameters: ion spray voltage, −4500 V (negative ion mode) and 5500 V (positive ion mode); collision gas, medium; Turbo V spray temperature, 600 °C; heater gas (Gas 2), 60 psi; nebulizer gas (Gas 1), 50 psi; curtain gas, 20 psi. Data was acquired using the multiple reaction monitoring (MRM) mode, and the retention time corresponding to each MRM transition was automatically set using the MassLynx software. The Peak View Software 2.2 (SCIEX, Foster City, CA, USA) was used for data analysis.

### 4.9. Ferulic Acid Analysis

#### 4.9.1. Sample Preparation

Ground *A. sinensis* roots (200 mg) were dissolved in 20 mL of 70% methanol, and the samples were weighed. The conical flasks were heated to reflux at 80 °C for 3 min. Then, the samples were cooled and weighed again, and 70% methanol was added to the flasks to compensate for the lost weight. After shaking the samples and allowing them to stand still, the supernatant was collected and filtered. The filtrate (10 mL) was passed through a 0.45 μm organic membrane. Ferulic acid in each sample was analyzed by high performance liquid chromatography (HPLC).

#### 4.9.2. HPLC Analysis

To conduct the HPLC analysis, standard ferulic acid (Batch number 17092501) was purchased from Chengdu Pufeide Biotechnology Co. Ltd., which was diluted to a concentration of 0.012 mg/mL using methanol. Quantitative and qualitative analyses of ferulic acid were performed using the Agilent 1260 (Agilent USA) Infinity II Quaternary Pump (G7111A). Data acquisition was performed using the Agilent HPLC software. Ferulic acid was separated on a unitary C18 chromatographic column (4.6 mm × 250 mm, 5 µm). Octadecyl silane bonded silica gel was used as a filler. The HPLC analysis was conducted using acetonitrile: 0.085%/phosphoric acid (20:80) solution as the mobile phase, with the following parameters: detection wavelength = 316 nm, column temperature = 35 °C, flow rate = 1 mL/min, injection volume = 10 μL. The calibration curve was constructed using serial dilutions of ferulic acid standard in methanol, with concentration ranging from 0.00324 to 0.054 mg/mL.

### 4.10. Competition Indices

#### 4.10.1. Land Equivalent Ratio (LER)

The LER index was adopted to compare the *A. sinensis*/*V. faba* intercropping system with the sole cropping system of each crop. The value of LER was calculated according to the following equation:
(1)$$\mathrm{LER}=\mathrm{LERv}+\mathrm{LERa}\;\mathrm{LERv}=\frac{\mathrm{Yvi}}{\mathrm{Yvm}}\;\mathrm{LERv}=\frac{\mathrm{Yvi}}{\mathrm{Yvm}}$$
where Yvm and Yam represent the yield of *V. faba* and *A. sinensis*, respectively, in the sole cropping system, and Yvi and Yai represent the yield of *V. faba* and *A. sinensis*, respectively, in the intercropping system.

LER > 1.0 indicates a yield advantage of intercropping over monoculture, whereas LER < 1.0 indicates no yield advantage of intercropping over monoculture [70,71].

#### 4.10.2. Aggressivity (A)

The competitive relationship between two crops was evaluated by the aggressivity (A) as proposed by Willey (1979) [72] using the following formula:(2)$$\mathrm{Aa}=\left(\frac{\mathrm{Yai}}{\mathrm{Yam}\times \mathrm{Zai}}\right)-\left(\frac{\mathrm{Yvi}}{\mathrm{Yvm}\times \mathrm{Zvi}}\right)\;\mathrm{Av}=\left(\frac{\mathrm{Yvi}}{\mathrm{Yvm}\times \mathrm{Zvi}}\right)-\left(\frac{\mathrm{Yai}}{\mathrm{Yam}\times \mathrm{Zai}}\right)$$
where Zai is the sown proportion of *A. sinensis* in intercropping with *V. faba*; and Zvi is the sown proportion of *V. faba* in intercropping with *A. sinensis*.

#### 4.10.3. Crowding Coefficient (RCC)

In the intercropping system, the relative dominance of one crop over the other was calculated with the crowding coefficient based on the following equation:
(3)$$\mathrm{RCC}=\mathrm{RCCv}+\mathrm{RCCa}\;\mathrm{RCCv}=\frac{\mathrm{Yvi}\times \mathrm{Zai}}{\left(\mathrm{Yvm}-\mathrm{Yvi}\right)\times \mathrm{Zvi}}\;\mathrm{RCCa}=\frac{\mathrm{Yai}\times \mathrm{Zvi}}{\left(\mathrm{Yam}-\mathrm{Yai}\right)\times \mathrm{Zai}}$$

When the values of the two coefficients (RCCv and RCCa) are >1, there is a yield advantage in intercropping as compared with monoculture; when they are equal to 1, there is no yield advantage, and when they are <1, there is a disadvantage in yield.

#### 4.10.4. Competitive Ratio (CR)

Another index, the competitive ratio (CR), was used to evaluate competitive ability among intercropping components. This index was more advantageous than other indices due to the fact that the CR gives stronger competitive ability to the species. The CR implies the ratio of the individual LER of the intercrop component in which they were initially sown proportionally. We used the following equation suggested by Dhima (2007) [37] to calculate the CR index:(4)$$\mathrm{CRv}=\left(\frac{\mathrm{LERv}}{\mathrm{LERa}}\right)\times \left(\frac{\mathrm{Zai}}{\mathrm{Zvi}}\right)\;\mathrm{CRa}=\left(\frac{\mathrm{LERa}}{\mathrm{LERv}}\right)\times \left(\frac{\mathrm{Zvi}}{\mathrm{Zai}}\right)$$
where CRv is the crowding ratio of *V. faba* and CRa is the crowding ratio of *A. sinensis*.

### 4.11. Statistical Analysis

All statistical analyses were performed using SPSS 21.0 (SPSS inc., Chicago, IL, USA). Prior to all statistical analyses, we tested the heterogeneity of variances using Levene’s test, and the original data were normalized using log transformation or standardization prior to statistical analysis when necessary. A two-AVOVA with post hoc tests was used to determine the effect of treatment, year, and treatment × year, on all variables, including soil properties (organic matter, available nitrogen, available phosphorus, and available potassium), early bolting rate, chlorophyll content, biomass, ferulic acid content, essential oil content, essential oil yield, and essential oil composition. Then, the R software (version 4.1.0) was used for the data analysis and production of figures. We used the packages “corrplot” and “psych” in the R software to calculate correlation coefficient and *p*-value of significance test.

## 5. Conclusions

Based on the outcomes of the field cultivation of *A. sinensis* using an intercropping system without fertilization, we reached the following conclusions as described below:

(1)Intercropping significantly reduces the early bolting rate of *A. sinensis.*(2)The AS/VF (3:1) intercropping pattern results in an optimal yield and the highest ferulic acid content of *A. sinensis*, highest biomass yield of *V. faba,* and highest land equivalent ratio (LER), thus, the AS/VF (3:1) intercropping pattern without fertilization is the most productive with high quality. These data indicate that intercropping can serve as an alternative for reducing the use of chemical fertilizers and can also decrease the early bolting rate of *A. sinensis*, thus, enabling its sustainable production.

## Figures and Tables

**Figure 1 plants-11-02950-f001:**
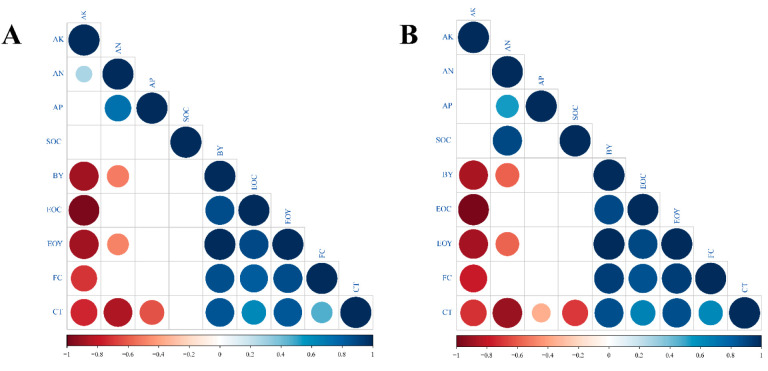
Correlations among soil organic matter, available N, P, and K, and biomass yield, ferulic acid content, essential oil content, essential oil yield: (**A**) in 2018; (**B**) in 2019 (Note: The criterion for statistical significance was set at *p* < 0.05; spaces indicate no correlation or insignificant correlation; AK represents available K; AN represents available K; AP represents available K; SOC represents soil organic carbon; BY represents biomass yield of *A. sinensis*; EOC represents essential oil content of *A. sinensis*; EOY represents essential oil yield of *A. sinensis*; FC represents ferulic acid content of *A. sinensis*; CT represents chloroplast content of *A. sinensis.*).

**Figure 2 plants-11-02950-f002:**
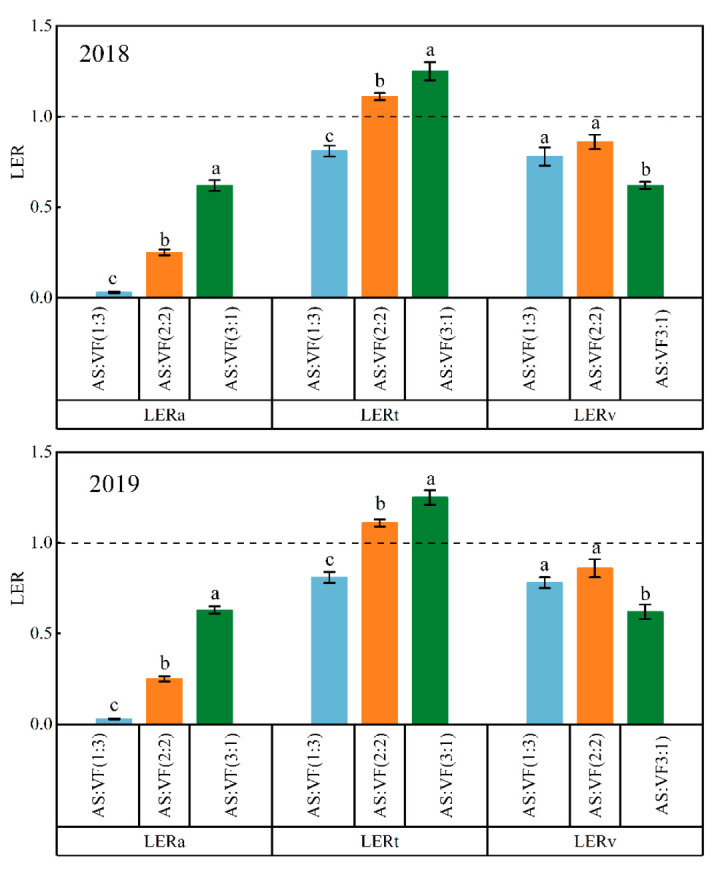
Land equivalent ratios (LERs) of the *A. sinensis*/*V. faba* intercropping patterns in 2018 and in 2019. Bars bearing different letters within the same LER indicate significance at *p* < 0.05. Error bars epitomize the standard deviation of the means. The dashed lines denote an LER equal to 1. AS/VF (3:1), three rows of *A. sinensis* + one row *V. faba*; AS/VF (2:2), two rows of *A. sinensis* + two rows *V. faba*; AS/VF (1:3), one row of *A. sinensis* + three rows of *V. faba*.

**Figure 3 plants-11-02950-f003:**
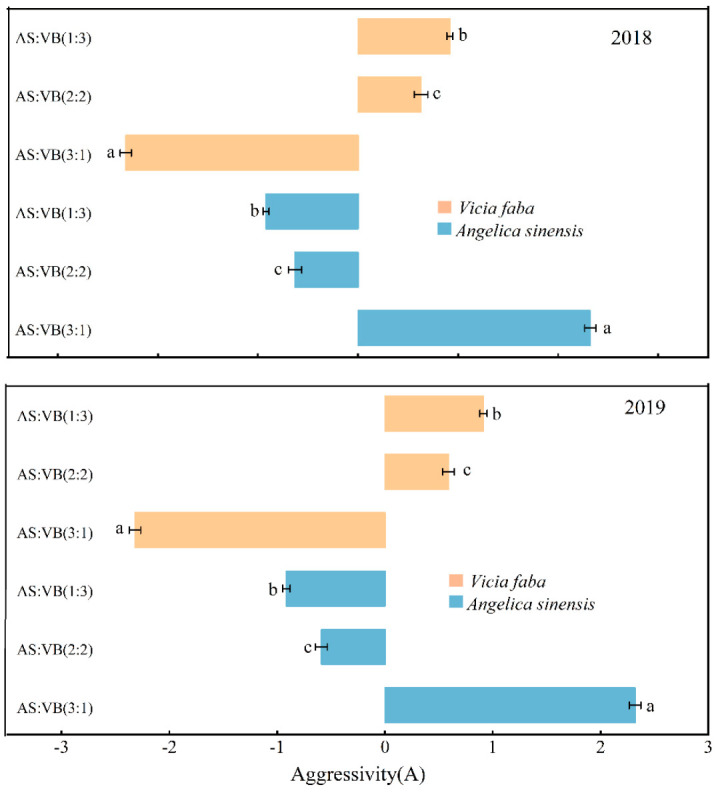
Aggressivity between *A. sinensis* and *V. faba* over 2018 and 2019 under the intercropping systems. Within a year, per companion crop, bars bearing the different letter are significantly diferent at *p* < 0.05. Error bars represent the standard deviation of the means. AS/VF (3:1), three rows of *A. sinensis* + one row *V. faba*; AS/VF (2:2), two rows of *A. sinensis* + two rows *V. faba*; AS/VF (1:3), one row of *A. sinensis* + three rows of *V. faba*.

**Figure 4 plants-11-02950-f004:**
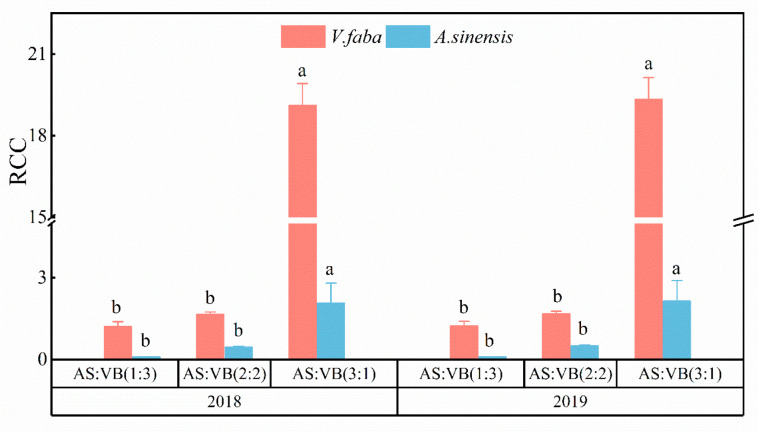
Crowding coefficient (RCC) values in different intercropping patterns in 2018 and in 2019. Within a year, per companion crop, bars bearing the same letter are significantly different at *p* < 0.05. Error bars represent the standard deviation of the means. AS/VF (3:1), three rows of *A. sinensis* + one row *V. faba*; AS/VF (2:2), two rows of *A. sinensis* + two rows *V. faba*; AS/VF (1:3), one row of *A. sinensis* + three rows of *V. faba*.

**Figure 5 plants-11-02950-f005:**
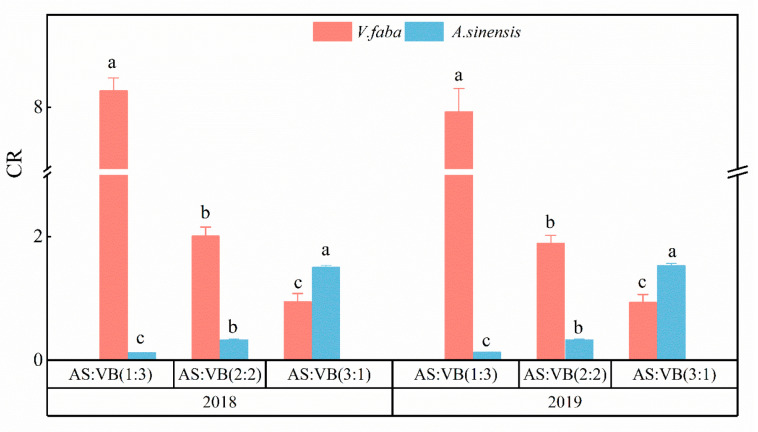
Competitive ratio (CR) values in different intercropping patterns in 2018 and in 2019. Within a year, per companion crop, bars bearing the same letter are significantly different at *p* < 0.05. Error bars represent the standard deviation of the means. AS/VF (3:1), three rows of *A. sinensis* + one row *V. faba*; AS/VF (2:2), two rows of *A. sinensis* + two rows *V. faba*; AS/VF (1:3), one row of *A. sinensis* + three rows of *V. faba*.

**Figure 6 plants-11-02950-f006:**
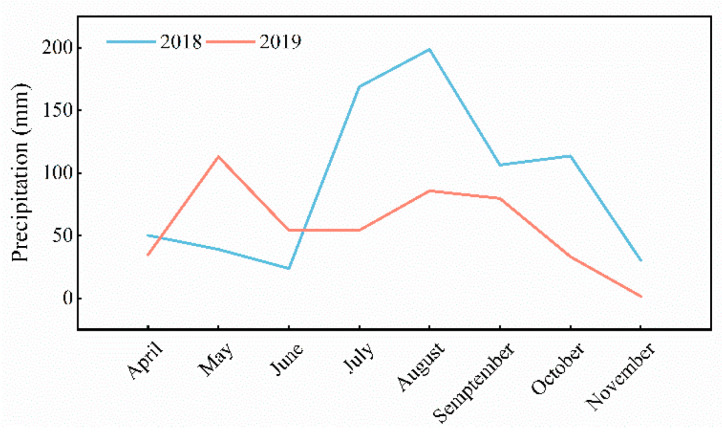
Average monthly precipitation (mm) of the experimental site during the summer in 2018 and in 2019.

**Table 1 plants-11-02950-t001:** Effects of cropping systems on soil characteristics.

Year	Treatments	Organic Matter (g/kg)	Available N (mg/kg)	Available P (mg/kg)	Available K (mg/kg)
2018	AS	16.67 ± 2.18 ^a^	61.58 ± 5.73 ^b^	19.19 ± 1.44 ^b^	100.67 ± 2.08 ^a^
	AS/VF (3:1)	18.54 ± 2.91 ^a^	74.17 ± 3.95 ^ab^	22.78 ± 0.34 ^a^	104.33 ± 8.02 ^a^
	AS/VF (2:2)	19.32 ± 3.11 ^a^	75.17 ± 3.47 ^a^	20.44 ± 1.05 ^ab^	103.00 ± 6.24 ^a^
	AS/VF (1:3)	16.30 ± 1.88 ^a^	71.60 ± 11.97 ^ab^	21.04 ± 2.66 ^ab^	118.00 ± 22.91 ^a^
2019	AS	16.43 ± 3.82 ^a^	61.25 ± 9.24 ^a^	18.85 ± 2.91 ^a^	100.33 ± 6.5 ^a^
	AS/VF (3:1)	17.96 ± 5.52 ^a^	73.50 ± 20.46 ^a^	22.92 ± 3.27 ^a^	103.67 ± 15.31 ^a^
	AS/VF (2:2)	18.63 ± 4.22 ^a^	73.50 ± 6.37 ^a^	19.11 ± 3.17 ^a^	102.00 ± 7.94 ^a^
	AS/VF (1:3)	17.46 ± 3.81 ^a^	72.27 ± 12.91 ^a^	20.35 ± 3.73 ^a^	117.33 ± 23.8 ^a^
Intercropping		*p* > 0.05	*p* > 0.05	*p* > 0.05	*p* > 0.05
Year		*p* > 0.05	*p* > 0.05	*p* > 0.05	*p* > 0.05
Intercropping × Year		*p* = 0.9552	*p* = 0.9968	*p* = 0.9669	*p* = 1.0000

Data are means of three triplicates ± SD. Means in a row with different superscripts differ significantly (*p* < 0.05, two-way ANOVA, Tukey-HSD). AS, sole cropping of *A. sinensis*; AS/VF (3:1), three rows of *A. sinensis* + one row *V. faba*; AS/VF (2:2), two rows of *A. sinensis* + two rows *V. faba*; AS/VF (1:3), one row of *A. sinensis* + three rows of *V. faba*.

**Table 2 plants-11-02950-t002:** Early bolting rate of *A. sinensis* under different cropping systems in 2018 and in 2019 ^1^.

Year	Treatments	Early Bolting Rate (%)
2018	AS	36.78 ± 0.48 ^a^
	AS/VF (3:1)	17.37 ± 0.56 ^d^
	AS/VF (2:2)	22.58 ± 1.01 ^c^
	AS/VF (1:3)	26.03 ± 0.26 ^b^
2019	AS	35.48 ± 0.54 ^a^
	AS/VF (3:1)	16.32 ± 1.23 ^d^
	AS/VF (2:2)	20.72 ± 0.56 ^c^
	AS/VF (1:3)	24.70 ± 1.22 ^b^
Intercropping		*p* < 0.05
Year		*p* < 0.05
Intercropping × Year		*p* = 0.8448

^1^ Data are means of three triplicates ± SD. Means in a row with different superscripts differ significantly (*p* < 0.05, two-way ANOVA, Tukey-HSD). AS, sole cropping of *A. sinensis*; AS/VF (3:1), three rows of *A. sinensis* + one row *V. faba*; AS/VF (2:2), two rows of *A. sinensis* + two rows *V. faba*; AS/VF (1:3), one row of *A. sinensis* + three rows of *V. faba*.

**Table 3 plants-11-02950-t003:** Chlorophyll contents and biomass yields of *A. sinensis* and *V. faba* under different cropping systems in 2018 and in 2019 ^1^.

Year	Treatments	Chlorophyll Contents		Biomass Yield(kg/hm^2^)	
		*A. sinensis*	*V. faba*	*A. sinensis*	*V. faba*
2018	AS	34.24 ± 0.57 ^a^		5024.8 ± 34.6 ^a^	
	VF		30.15 ± 2.55 ^c^		4319.6 ± 167.0 ^d^
	AS/VF (3:1)	30.56 ± 2.97 ^a^	50.33 ± 1.22 ^a^	4196.5 ± 34.8 ^b^	10763.5 ± 542.9 ^a^
	AS/VF (2:2)	29.86 ± 2.41 ^a^	42.13 ± 2.13 ^b^	2520.4 ± 72.4 ^c^	7465.6 ± 232.1 ^b^
	AS/VF (1:3)	28.26 ± 2.01 ^a^	30.19 ± 2.43 ^c^	525.0 ± 9.0 ^d^	4510.2 ± 435.3 ^c^
2019	AS	36.89 ± 1.26 ^a^		5116.5 ± 145.5 ^a^	
	VF		33.54 ± 1.68 ^c^		4433.3 ± 146.8 ^d^
	AS/VF (3:1)	32.78 ± 1.56 ^a^	49.68 ± 1.56 ^a^	4284.0 ± 150.5 ^b^	11110.8 ± 516.5 ^a^
	AS/VF (2:2)	31.69 ± 0.99 ^a^	41.38 ± 1.33 ^b^	2618.9 ± 81.0 ^c^	7633.3 ± 148.4 ^b^
	AS/VF (1:3)	30.77 ± 1.33 ^a^	29.66 ± 1.89 ^c^	638.5 ± 19.9 ^d^	4625.9 ± 25.0 ^c^
Intercropping		*p* > 0.05	*p* < 0.05	*p* < 0.05	*p* < 0.05
Year		*p* > 0.05	*p* > 0.05	*p* > 0.05	*p* > 0.05
Intercropping × Year		*p* = 0.9986	*p* = 0.112	*p* = 0.9678	*p* = 0.9918

^1^ Data are means of three triplicates ± SD. Means in a row with different superscripts differ significantly (*p* < 0.05, two-way ANOVA, Tukey-HSD). AS, sole cropping of *A. sinensis*; VF, sole cropping of *V. faba*; AS/VF (3:1), three rows of *A. sinensis* + one row *V. faba*; AS/VF (2:2), two rows of *A. sinensis* + two rows *V. faba*; AS/VF (1:3), one row of *A. sinensis* + three rows of *V. faba*.

**Table 4 plants-11-02950-t004:** Ferulic acid content, essential oil content, and yield of *A. sinensis* under different cropping systems in 2018 and in 2019 ^1^.

Year	Treatments	Ferulic Acid Content (%)	Essential Oil Content (%)	Essential Oil Yield (kg/hm^2^)
2018	AS	0.139 ± 0.0020 ^ab^	0.946 ± 0.0083 ^a^	47.70 ± 1.43 ^a^
	AS/VF (3:1)	0.160 ± 0.0039 ^a^	0.9267 ± 0.0058 ^a^	40.34 ± 1.65 ^b^
	AS/VF (2:2)	0.121 ± 0.0018 ^ab^	0.961 ± 0.0031 ^a^	23.36 ± 0.81 ^c^
	AS/VF (1:3)	0.101 ± 0.0053 ^b^	0.502 ± 0.0018 ^b^	2.63 ± 0.12 ^d^
2019	AS	0.1332 ± 0.002 ^ab^	0.916 ± 0.086 ^a^	46.89 ± 1.28 ^a^
	AS/VF (3:1)	0.145 ± 0.0018 ^a^	0.884 ± 0.063 ^a^	39.04 ± 1.20 ^b^
	AS/VF (2:2)	0.115 ± 0.0018 ^ab^	0.923 ± 0.031 ^a^	23.20 ± 1.39 ^c^
	AS/VF (1:3)	0.0959 ± 0.0053 ^b^	0.452 ± 0.017 ^b^	2.88 ± 0.10 ^d^
Intercropping		*p* < 0.05	*p* < 0.05	*p* < 0.05
Year		*p* > 0.05	*p* > 0.05	*p* > 0.05
Intercropping × Year		*p* = 0.9699	*p* = 0.9843	*p* = 0.9678

^1^ Data are means of three triplicates ± SD. Means in a row with different superscripts differ significantly (*p* < 0.05, two-way ANOVA, Tukey-HSD). AS, sole cropping of *A. sinensis*; AS/VF (3:1), three rows of *A. sinensis* + one row *V. faba*; AS/VF (2:2), two rows of *A. sinensis* + two rows *V. faba*; AS/VF (1:3), one row of *A. sinensis* + three rows of *V. faba*.

**Table 5 plants-11-02950-t005:** Essential oil compositions of *A. sinensis* under different cropping systems in 2018 and in 2019 ^1^.

Year	Treatments	Ligustilide (%)	Senkyunolide A (%)	Senkyunolide I (%)	Senkyunolide H (%)	Btylphthaide (%)	Butylidenephalide (%)	Levistolide A (%)
2018	AS	15.07 ± 2.80 ^a^	0.48 ± 0.089 ^a^	0.21 ± 0.061 ^ab^	0.042 ± 0.0011 ^ab^	0.036 ± 0.0087 ^a^	0.16 ± 0.023 ^a^	0.035 ± 0.0048 ^b^
	AS/VF(3:1)	12.38 ± 1.27 ^a^	0.30 ± 0.013 ^a^	0.15 ± 0.018 ^b^	0.027 ± 0.0021 ^b^	0.019 ± 0.0033 ^a^	0.12 ± 0.017 ^ab^	0.032 ± 0.0014 ^b^
	AS/VF(2:2)	15.25 ± 1.27 ^a^	0.406 ± 0.077 ^a^	0.20 ± 0.033 ^ab^	0.037 ± 0.0029 ^b^	0.036 ± 0.0011 ^a^	0.067 ± 0.026 ^b^	0.057 ± 0.0077 ^a^
	AS/VF(1:3)	15.10 ± 3.78 ^a^	0.376 ± 0.031 ^a^	0.25 ± 0.063 ^a^	0.056 ± 0.0011 ^a^	0.034 ± 0.0017 ^a^	0.14 ± 0.039 ^ab^	0.050 ± 0.0013 ^a^
2019	AS	14.46 ± 3.40 ^a^	0.46 ± 0.075 ^a^	0.22 ± 0.077 ^ab^	0.045 ± 0.0016 ^ab^	0.043 ± 0.002 ^a^	0.16 ± 0.031 ^a^	0.038 ± 0.0060 ^b^
	AS/VF(3:1)	13.04 ± 2.26 ^a^	0.33 ± 0.019 ^a^	0.14 ± 0.028 ^b^	0.026 ± 0.0033 ^b^	0.023 ± 0.0074 ^a^	0.11 ± 0.026 ^ab^	0.035 ± 0.0067 ^b^
	AS/VF(2:2)	15.58 ± 1.84 ^a^	0.396 ± 0.094 ^a^	0.19 ± 0.038 ^ab^	0.036 ± 0.0039 ^b^	0.039 ± 0.0016 ^a^	0.064 ± 0.031 ^b^	0.054 ± 0.0013 ^a^
	AS/VF(1:3)	14.44 ± 4.5 ^a^	0.409 ± 0.037 ^a^	0.25 ± 0.094 ^a^	0.054 ± 0.0015 ^a^	0.032 ± 0.0019 ^a^	0.12 ± 0.039 ^ab^	0.047 ± 0.0016 ^a^
Intercroping		*p* > 0.05	*p* > 0.05	*p* < 0.05	*p* < 0.05	*p* > 0.05	*p* < 0.05	*p* < 0.05
Year		*p* > 0.05	*p* > 0.05	*p* > 0.05	*p* > 0.05	*p* > 0.05	*p* > 0.05	*p* > 0.05
Intercropping × Year		*p* = 0.9689	*p* = 0.9914	*p* = 0.9925	*p* = 0.9715	*p* = 0.9245	*p* = 0.9782	*p* = 0.8958

^1^ Data are means of three triplicates ± SD. Means in a row with different superscripts differ significantly (*p* < 0.05, two-way ANOVA, Tukey-HSD). AS, sole cropping of *A. sinensis*; AS/VF (3:1), three rows of *A. sinensis* + one row *V. faba*; AS/VF (2:2), two rows of *A. sinensis* + two rows *V. faba*; AS/VF (1:3), one row of *A. sinensis* + three rows of *V. faba*.

## Data Availability

Not applicable.

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
