# Peer review of "Early Bolting, Yield, and Quality of Angelica sinensis (Oliv.) Diels Responses to Intercropping Patterns"

_plants, 2022, doi:10.3390/plants11212950_

Round 1

Reviewer 1 Report

1. Lack of detailed information on the applied statistical analyzes. Whether or not their basic assumptions are met (homogeneity of variance, normal distribution). What post-hoc tests were used (complete e.g. for Table 1)? Was standard or non-parametric ANOVA  performed? Have "years" been taken into account and  in which analyzes? Was it fix or  random factor?

2. Correlations should be determined separately in years. The lack of significance for coefficients greater (in absolute value) than 0.8 indicates a very small sample. What is its size. If the mean over the years is used, what is the basis for this.

3. This type of research should be supplemented with competition models.

4. The abstract cosists the cultivation of soybean.

5. The content of the paragraph starting with line 87 is the repetition of the content of the first paragraph of the introduction (from line 34).

Reviewer 2 Report

The idea of ​​the experiment is very interesting and economically important. Describing the results is extremely careless. I have listed only the most obvious errors below.

Title: Why “improves”? It should be “lowers”, “reduces” or “decreased”.

Abstract: Why are A. sinansis and Vicia faba not highlighted in italics? Why the authors use the term soybean? This is a completely different plant than a faba bean.

Line 24: Is “highest grain yield of soybean” (faba bean?) The yield of dry biomass, not grain, was determined. By the way “grain” is correct for cereals (Poaceae family), the yields of other plants are seeds.

Line 34: The name Umbelliferae is no longer used. Now we use Apiaceae.

Lines 37-42: The same text is repeated in lines 87-92.

Line 108: The second hypothesis is justifiable because there was no fertilization treatment.

Line 120: Why “of Angelica sinensis”?  Moreover “in 2018 and 2019” should be deleted because it is in the table.

Lines 122 and 123: The comment on the data in the table 2 is not correct.

In figures 2B and 3b is AS (monoculture) but there should be FB. Why are uppercase and lowercase letters used interchangeably to denote figures?

Figure 5B. It is not possible for the yield of essential oil per hectare to be 4500 kg. It was certainly 45 kg per hectare.

Lines 249-250: “Therefore, the drought in 2018 led to the high early bolting rate of A. sinensis.

This is an overinterpretation as the differences were within the limits of the statistical error.

Methods

It is a pity that the shading of A. sinensis plants was not determined.

Why were the results computed statistically for each year separately and not as a two-year synthesis?

Round 2

Reviewer 2 Report

The authors revised the manuscript as suggested and explained my doubts.

I like the  title proposed “Early bolting rate, yield and quality of Angelica sinensis (Oliv.) Diels as response to intercropping system”. It  sounds much better than the current title.

However I  still see typing errors e.g.
line 33: is “V.faba”, should be “V. faba” or better “Vicia faba” (italics)

Line 33: is “Total”, should be “total”

Abbreviations used in the abstract should be explained: “A” in the line 24 and “CR” in the line 25.

In the whole text is: “Table 1” should be “Table 1.” And the same with Figures.

V. faba is written in different ways:  without space or with a space, italic or regular font.
